biochemistry

graphene quantum dots, β-amyloid aggregation, modulation

**Authors for correspondence:**
Yanlian Yang
e-mail: yangyl@nanoctr.cn
Chen Wang
e-mail: wangch@nanoctr.cn

This article has been edited by the Royal Society of Chemistry, including the commissioning, peer review process and editorial aspects up to the point of acceptance.

# Modulation of β-amyloid aggregation by graphene quantum dots

Changliang Liu[1,2], Huan Huang[1,2], Lilusi Ma[1], Xiaocui Fang[1], Chen Wang[1,2] and Yanlian Yang[1,2]

[1]CAS Key Laboratory of Standardization and Measurement for Nanotechnology, CAS Key Laboratory of Biological Effects of Nanomaterials and Nanosafety, CAS Center for Excellence in Nanoscience, National Center for Nanoscience and Technology, Beijing 100190, People's Republic of China
[2]University of Chinese Academy of Sciences, Beijing 100049, People's Republic of China

CL, 0000-0002-9905-5428

Misfolding and abnormal aggregation of β-amyloid peptide is associated with the onset and progress of Alzheimer's disease (AD). Therefore, modulating β-amyloid aggregation is critical for the treatment of AD. Herein, we studied the regulatory effects and mechanism of graphene quantum dots (GQDs) on 1–42 β-amyloid ($A\beta_{1-42}$) aggregation. GQDs displayed significant regulatory effects on the aggregation of $A\beta_{1-42}$ peptide as detected by thioflavin T (ThT) assay. Then, the changes of confirmations and structures induced by GQDs on the $A\beta_{1-42}$ aggregation were monitored by circular dichroism (CD), dynamic light scattering (DLS) and transmission electron microscope (TEM). The *in vitro* cytotoxicity experiments further demonstrated the feasibility of GQDs on the regulation of $A\beta_{1-42}$ aggregation. Meanwhile, the structural changes of a $A\beta_{1-42}$/GQDs mixture in different pH revealed that electrostatic interaction was the major driving force in the co-assembly process of $A\beta_{1-42}$ and GQDs. The proposed mechanism of the regulatory effects of GQDs on the $A\beta_{1-42}$ aggregation was also deduced reasonably. This work not only demonstrated the potential feasibility of GQDs as therapeutic drug for AD but also clarified the regulatory mechanism of GQDs on the $A\beta_{1-42}$ aggregation.

## 1. Introduction

Misfolding and abnormal aggregation of amyloid peptides or proteins is associated with a large number of diseases, such as Alzheimer's disease (AD), Parkinson's disease (PD), Huntington's disease, type II diabetes, mad cow disease and some types of cancers [1–3]. Among which, AD is the most common neurodegenerative disease which is characterized by

the cerebral extracellular amyloid plaques [4,5]. The amyloid plaques are considered to be formed by the aggregation of β-amyloid peptide, which is an amphipathic polypeptide composed of 40–42 amino acids in humans [6]. Besides, $A\beta_{1-42}$ is the major variant in cerebrospinal fluid (CSF), and its aggregation is the main component of the amyloid plaques. Therefore, $A\beta_{1-42}$ peptide is considered as a major diagnostic biomarker and therapeutic target in the diagnosis and therapy of AD.

The sequence of $A\beta_{1-42}$ peptide has been determined to be 'AIVVGGVMLGIIAGKNSGVD EAFFVLKQHHVEYGSDHRFEAD' in humans, which contains some hydrophobic amino acid residues and positive charge residues. These amino acids play a key role in the aggregation of $A\beta_{1-42}$ driven by hydrophobic interaction, electrostatic interaction and hydrogen bond [7]. Aggregation of $A\beta_{1-42}$ would form soluble oligomers and mature fibrils, which would damage the neurocytes and then induce the cerebral degeneration due to the neurotoxicity of the aggregates [8]. Thus, inhibiting or eliminating the formation of toxic $A\beta_{1-42}$ aggregates would be an incredible strategy in AD therapy.

Recently, great efforts have been made to find efficient modulators to inhibit the toxic intermediates in the fibrillation process of amyloid peptides. Nanomaterials, including gold nanoparticles [9], inorganic semiconductors [10], polymer nanoparticles [11], magnetic nanoparticles [12], dendrimer [13] and carbon materials [14–22], have been intensely studied as new modulators in the field of amyloidosis with the development of nanotechnology. Notably, various carbon-based materials, such as graphene oxide (GO), carbon quantum dots and graphene quantum dots (GQDs), have drawn extensive attention due to their unique properties, versatile functionalities and good biocompatibility [23]. In particular, GQDs, as a zero-dimensional nanomaterial, possess superior chemical inertness, low toxicity, high fluorescent activity and excellent photostability [24]. Moreover, the unique structure and chemical properties make GQDs a research hotspot in the fields of imaging and sensing [25,26]. Several studies have reported that GQDs could be used to inhibit the aggregation of amyloid peptide [16–22], which provides the potentiality of GQDs in modulating amyloidogenesis. However, more characterization and studies were also needed to better understand the regulatory mechanism of GQDs for further application.

In this work, we investigated the modulation effects of GQDs on the aggregation of $A\beta_{1-42}$. Results indicated that the regulatory behaviour of GQDs was mass ratio dependent and the critical mass ratio of $A\beta_{1-42}$: GQDs was $1:1$. Above the critical mass ratio, GQDs could protect $A\beta_{1-42}$ peptide from aggregating into mature fibrils, instead of forming a hybrid network structure with β-turn confirmation with nontoxicity to cells according to the *in vitro* experiment. Then, the driven force of GQDs in the process of $A\beta_{1-42}$ aggregation was studied, and results indicated that the electrostatic interaction between GQDs and $A\beta_{1-42}$ peptide played an important role in the regulatory process. The regulation mechanism was further speculated according to the results in this work.

# 2. Material and methods

## 2.1. Materials

$A\beta_{1-42}$ peptide was synthesized by Science Peptide Biological Technology Co., Ltd. (Shanghai, China). Graphite powder (size less than 30 μm) and 1,1,1,3,3,3-hexafluoro-2-propanol (HFIP) were purchased from Sinopharm Chemical Reagent Co., Ltd. Concentrated sulfuric acid, sodium nitrate, potassium permanganate, hydrochloric acid, hydrogen peroxide, sodium hydroxide and sodium carbonate were purchased from Beijing Chemical Plant. MTT (3-(4,5-dimethyl-2-thiazolyl)-2,5-diphenyl-2-H-tetrazolium bromide) was bought from Solarbio Science & Technology Co., Ltd. (Beijing, China). Deionized water used in all experiments was obtained from Milli-Q Integral 3 (Merck Millipore, France).

## 2.2. Preparation of graphene quantum dots

GQDs were prepared through two step reactions. First, GO was prepared from graphite powder according to the Hummer method with minor modification [27]. In the typical experiment, 4 g graphite powder and 3 g $NaNO_3$ were dispersed into 150 ml concentrated $H_2SO_4$ under ice bath, then 18 g $KMnO_4$ was slowly added into the mixture with vigorous vortex. Next, the mixture was removed from the ice bath and reacted for 120 h at room temperature. The obtained brownish grey paste was washed with 3% $H_2O_2$, 1% HCl and deionized water successively. The crude product was dialysed against deionized water for 2 days followed by ultrasonic exfoliating for 2 h to get well-dispersed GO.

The GQDs were prepared by incision of the prepared GO through oxidation reaction [28]. Briefly, the GO was diluted into 5 mg ml$^{-1}$ with deionized water. Later, 30 ml GO solution, 1 ml concentrated $H_2SO_4$

and 1 ml $HNO_3$ were mixed together and then put into a microwave reactor for 2 h at 200°C with the power of 500 W. Subsequently, the product was cooled to room temperature and neutralized with $Na_2CO_3$; the supernatant was centrifuged at 10 000 rpm for 30 min to remove big particles. Last, the solution was filtered through a 0.22 µm microporous membrane and further dialysed (molecular weight cut-off: 1000 Da) against water for 3 days to obtain the final brown GQDs products.

## 2.3. Characterization of GQDs

The morphology of GQDs was performed on an FEI Tecnai 20 transmission electron microscope (TEM, USA) at an acceleration voltage of 200 kV. The PL spectra of GQDs were recorded on an F-4600 fluorescence spectrometer (Hitachi, Japan) with the excitation wavelength of 360, 400, 440, 500 and 520 nm at the concentration of 1 mg ml$^{-1}$. The emission spectra of GQDs with a series of concentrations were tested at the excitation wavelength of 360 nm. The structure of GQDs was further characterized by Raman spectra, Fourier-transform infrared spectra (FTIR) and X-ray photoelectron spectra (XPS). Specifically, Raman spectra were performed by irradiating laser at 514 nm in a Renishaw InVia Raman spectrometer (Renishaw plc, UK). FTIR was performed on a PerkinElmer FTIR spectrometer between 450 and 4000 cm$^{-1}$, and the sample disc was made by tableting the mixture of GQDs and KBr before the test. XPS was investigated on an ESCALab220i-XL electron spectrometer (VG Scientific Ltd, UK) with a high-performance Al monochromatic source of 15 kV.

## 2.4. Pre-treatment of A$\beta_{1-42}$ peptide

A$\beta_{1-42}$ powder was dissolved in 1,1,1,3,3,3-hexafluoro-2-propanol (HFIP) at a concentration of 1 mg ml$^{-1}$ and further incubated at 37°C for 6 h. The HFIP could be used as a hydrogen bond breaker to eliminate pre-existing homogeneity structures in the A$\beta_{1-42}$. The A$\beta_{1-42}$ solution was then separated into 100 µl per tube and stored at 4°C as stock solutions of A$\beta_{1-42}$ peptide.

## 2.5. ThT assay

One hundred microlitre stock solution of A$\beta_{1-42}$ peptide pre-treated as above was taken out and dried under nitrogen gas. The A$\beta_{1-42}$ peptide was then re-dissolved into PBS buffer at a concentration of 50 µM with 10 µl DMSO as cosolvent. Thioflavin T (ThT) stock solution was prepared by dissolving ThT powder into PBS at a concentration of 1 mM and filtered by a 0.22 µm filter membrane before use. For each measurement, the incubated A$\beta_{1-42}$ solutions at a concentration of 20 µM were prepared with ThT solution in phosphate buffer and GQDs at a mass ratio of 100 : 1, 10 : 1, 1 : 1 and 1 : 5. The final concentration of ThT was 10 µM, and free A$\beta_{1-42}$ peptide was used as the control. The ThT fluorescence intensity was recorded every 10 min on a microplate reader (Tecan infinite M200, Switzerland) at excitation and emission wavelength of 450 nm and 482 nm, respectively.

## 2.6. Circular dichroism spectroscopy assay

To investigate the influences of GQDs in the aggregation process of A$\beta_{1-42}$, changes of the secondary structure of A$\beta_{1-42}$ with different amounts of GQDs were measured by CD analysis on a J-1500 spectropolarimeter (Jasco, Japan) at room temperature under a constant flow of nitrogen gas. Briefly, the A$\beta_{1-42}$ peptides were dissolved in PB buffer (pH = 7.0) with 1% acetonitrile, then GQDs were added into the A$\beta_{1-42}$ peptide solution with the mass ratio of A$\beta_{1-42}$: GQDs in 100 : 1, 10 : 1, 1 : 1 and 1 : 5. Free A$\beta_{1-42}$ peptide was prepared with the same steps mentioned before as control group. The samples were incubated in a shaker at 37°C before detection. Later, the samples were loaded into the CD cuvette to record CD spectra at 0 h, 2 h, 4 h, 6 h, 8 h and 10 h, respectively. GQDs solution with the same condition of samples was used as baseline in the experiment.

## 2.7. Cytotoxicity assay

Human neuroblastoma SH-SY5Y cells were cultured in Roswell Park Memorial Institute (RPMI)-1640 medium (Gibco, USA) supplemented with 10% fetal bovine serum (Gibco, USA) at 37°C in a humidified (5% $CO_2$, 95% air) incubator. SH-SY5Y cells were seeded into 96-well microplates with a density of 8000 cells per well in 100 µl complete medium. After being incubated overnight at 37°C, the cells were treated with different media containing A$\beta_{1-42}$ and A$\beta_{1-42}$/GQDs (mass ratio = 1 : 1)

mixtures. The concentration of $A\beta_{1-42}$ was in the range of 5–100 µM. After being incubated for another 72 h at 37°C, 20 µl MTT (5 mg ml$^{-1}$ of MTT in Hank's balancing buffer) was added into each well and the cells were incubated for another 4 h. Then, the medium was aspirated and 100 µl DMSO was added to dissolve the formazan salt. Cell viability was expressed as the percentage of viable cell relative to untreated group using the absorbance at 490 nm by an Infinite 200 microplate reader, TECAN. The cytotoxicity of GQDs was detected by the same procedure, and the concentration of GQDs was in the range of 22.5–1200 µg ml$^{-1}$.

## 2.8. Dynamic light scattering assay

To detect the regulatory effect of GQDs in the process of $A\beta_{1-42}$ aggregation, DLS tests were performed on a ZetasizerNano ZS nanoparticle size analyser (Malvern Instruments Ltd, UK). The $A\beta_{1-42}$/GQDs mixture with the mass ratio of 1:1 was prepared in PBS buffer (pH = 7.0) first, and the final concentration of $A\beta_{1-42}$ was 20 µM. After being incubated for 72 h at 37°C, the mixture was transferred into the DLS instrument to measure the size distribution of the peptides. Another sample was prepared with the same procedure mentioned above. At 72 h, the pH value of the mixture was adjusted to 10 by dropping NaOH solution (0.1 M) into the system. After 72 h incubation, the size distribution of the mixture was measured at the same condition.

The regulatory effect of GQDs for the $A\beta_{1-42}$ fibrils was further investigated by the DLS assay. $A\beta_{1-42}$ fibrils were made by dissolving the $A\beta_{1-42}$ peptide into PBS (pH = 7.0) at a concentration of 20 µM and incubating for 72 h at 37°C. Then, the GQDs were added into the $A\beta_{1-42}$ fibrils with the mass ratio of 1:1. DLS was performed to detect the changes in the size distribution of $A\beta_{1-42}$ fibrils after the mixture being incubated for 72 h. Later, the value of pH was adjusted to 10 by 0.1 M NaOH solution and incubated for another 72 h, and the same procedure was performed to detect the size change of the $A\beta_{1-42}$ fibrils.

## 2.9. Transmission electron microscope

First, the morphology of the $A\beta_{1-42}$ aggregates with different contents of GQDs was determined by TEM. The samples were made by mixing the $A\beta_{1-42}$ peptide and GQDs in PBS (pH = 7.0) with the mass ratio of 100:1 and 1:1 ($A\beta_{1-42}$:GQDs). After being incubated for 72 h at 37°C, the samples were dropped on the carbon-coated grid and observed on an Ht-7700 TEM (Hitachi, Japan) with the accelerate voltage of 80 kV. Then, we chose the mass ratio of 1:1 for $A\beta_{1-42}$:GQDs to investigate the regulatory effect of GQDs in the $A\beta_{1-42}$ aggregation. The samples were prepared by incubating the $A\beta_{1-42}$/GQDs mixture for 72 h before observation. Then, the pH value of the $A\beta_{1-42}$/GQDs mixture was adjusted to 10 by NaOH solution (0.1 M) and reacting for another 72 h before observation by TEM. Additionally, the regulatory effect of GQDs for the mature $A\beta_{1-42}$ fibrils was further investigated by TEM. Specifically, the samples were prepared by adding GQDs into the $A\beta_{1-42}$ mature fibrils with the mass ratio of 1:1, and the TEM was executed after 72 h of incubation. Then, the pH value was adjusted to 10 and the mixture was incubated for another 72 h before the TEM assay.

# 3. Results and discussion

## 3.1. Preparation and characterization of GQDs

GQDs were prepared through two steps. First, graphite powder was oxidized to GO according to the Hummer method [27]. Then, the GO was further oxidized to GQDs in the presence of concentrated $H_2SO_4$ and $HNO_3$. The prepared GQDs exhibited a sheet-like structure and monodisperse in aqueous according to the TEM images (figure 1a). Size distribution of the GQDs was measured and the statistics were processed by Nano Measurer Software 1.2.0 according to the TEM results, which revealed the lateral size as $4.0 \pm 0.7$ nm (electronic supplementary material, figure S1). Fluorescent properties were further investigated by PL spectra with different excitation wavelengths as shown in electronic supplementary material, figure S2. Results demonstrated an excitation-dependent behaviour of the emission spectra with the strongest PL peak shifting to longer wavelength when the excitation wavelength changed from 360 to 520 nm. Meanwhile, the strongest PL peak was nearly 500 nm with the excitation wavelength between 360 and 440 nm, and the maximum excitation wavelength was 360 nm. The emission spectra of GQDs also indicated a concentration-dependent PL behaviour at the

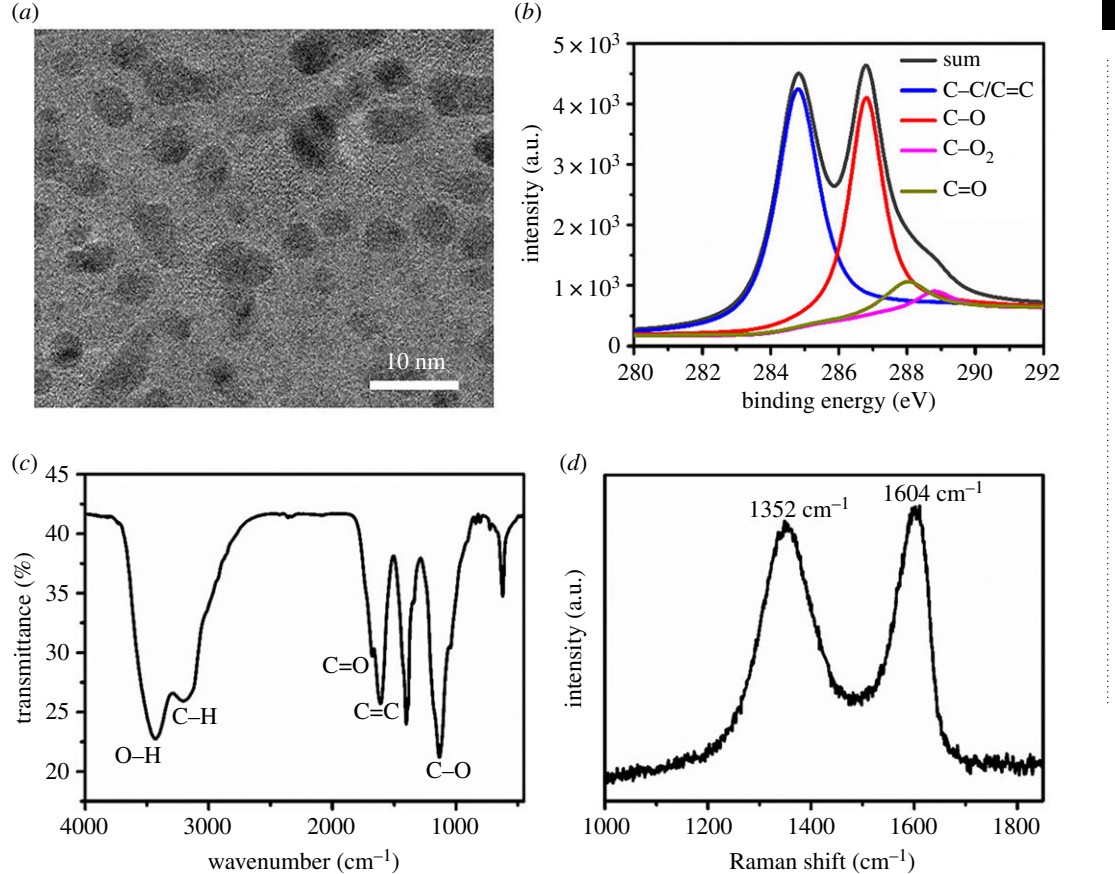

**Figure 1.** Morphology and spectroscopic characterizations of graphene quantum dots (GQDs). (a) TEM, scale bar is 10 nm; (b) XPS spectrum; (c) FTIR spectrum and (d) Raman spectrum.

excitation of 360 nm, and the fluorescent intensity was linearly correlated with the concentration at the range of 5–50 µg ml$^{-1}$ (electronic supplementary material, figure S3). The XPS (figure 1b) and FTIR (figure 1c) results demonstrated a high amount of oxygen element resulting from the formation of epoxide, hydroxyl, carbonyl and carboxyl in the oxidation process. The peaks of 1352 and 1604 cm$^{-1}$ in the Raman spectrum were in accordance with the D-band and G-band, which indicated the similar structure of GQDs with GO. The characterizations of GQDs revealed that the prepared GQDs not only have a conjugated structure like graphene but also have abundant oxygen-containing functional groups, which might contribute to the interactions between GQDs and Aβ$_{1-42}$ such as hydrophobic interaction, π–π stacking interaction, electrostatic interaction and hydrogen bond.

## 3.2. Regulatory effects of GQDs in the aggregation process of Aβ$_{1-42}$

As a material of regulator, how the GQDs interact with Aβ$_{1-42}$ peptide and what changes it brings to the aggregation process of Aβ$_{1-42}$ peptide are crucial questions. In this work, ThT assay, a widely used method for monitoring amyloid aggregation, was first performed to detect the aggregation kinetic process of Aβ$_{1-42}$ peptide in the presence of GQDs. The ThT molecules could fluoresce upon binding to the β-sheet structure specifically, which provides information on the nucleation and elongation phases of Aβ$_{1-42}$ assembly [29]. As shown in electronic supplementary material, figure S4, Aβ$_{1-42}$ peptide displayed a quasi-sigmoid binding curve characterized by a lag time of about 120 min, and about 600 min period of successively increasing ThT binding followed by a plateau after 600 min. The aggregation of Aβ$_{1-42}$ peptide was significantly different in the presence of GQDs compared with the free Aβ$_{1-42}$ peptide, indicating that GQDs could influence the aggregation of Aβ$_{1-42}$ peptide. Furthermore, CD spectra were performed to investigate the confirmational changes of Aβ$_{1-42}$ peptide on the secondary structure with or without GQDs. Figure 2 exhibited the variations of CD signals along with different amounts of GQDs at different time intervals. At 0 h, all the samples possessed the typical α-helix confirmation with a positive band nearly 195 nm and several weak negative bands

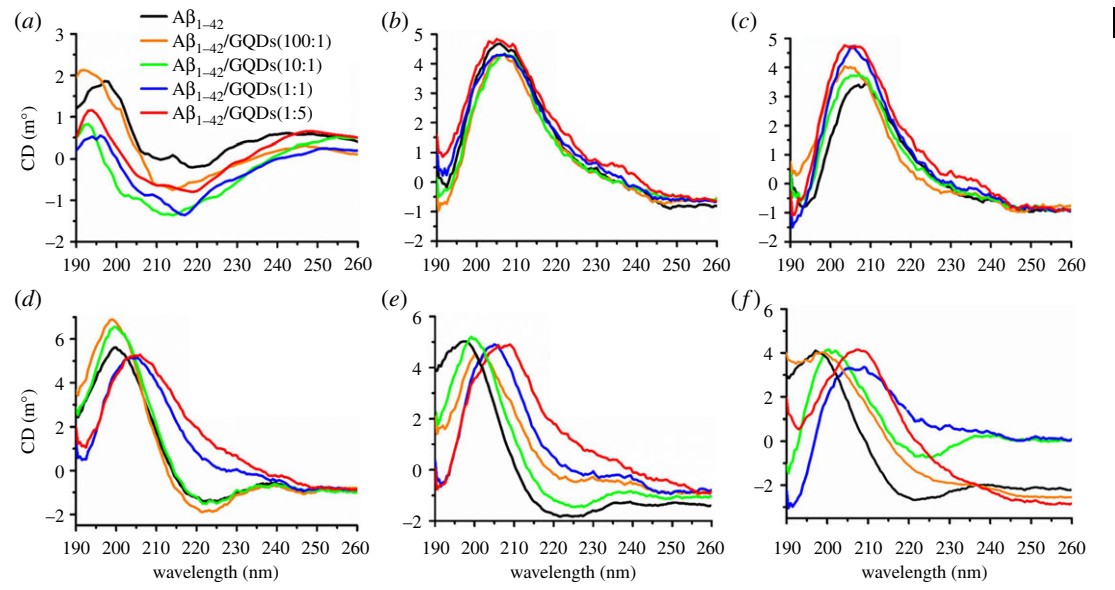

**Figure 2.** CD spectrum of Aβ$_{1-42}$ peptide and Aβ$_{1-42}$/GQDs mixture with different mass ratios (Aβ$_{1-42}$ : GQDs = 100 : 1, 10 : 1, 1 : 1 and 5 : 1) at (a) 0 h, (b) 2 h, (c) 4 h, (d) 6 h, (e) 8 h and (f) 10 h.

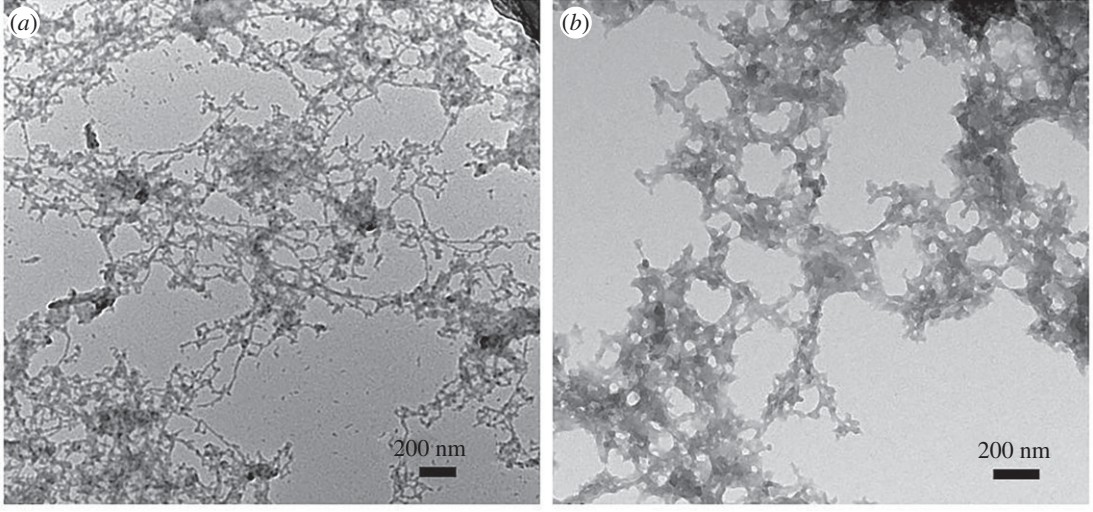

**Figure 3.** TEM images of the Aβ$_{1-42}$ peptide/GQDs mixture with different mass ratios: (a) Aβ$_{1-42}$ : GQDs = 100 : 1 and (b) Aβ$_{1-42}$ : GQDs = 1 : 1.

between 205 and 225 nm (figure 2a). After incubating for 2 h (figure 2b), both Aβ$_{1-42}$ peptide and Aβ$_{1-42}$/GQDs mixtures exhibited strong positive band at about 205 nm, which implied that the confirmation of Aβ$_{1-42}$ was changed from α-helix to β-turn. This secondary structure could maintain for 4 h (figure 2c). While after another 2 h of incubation (figure 2d), obvious changes occurred in some of these groups. Pure Aβ$_{1-42}$ peptide and Aβ$_{1-42}$/GQDs mixtures that have low content of GQDs (Aβ$_{1-42}$ : GQDs = 100 : 1 and 10 : 1) changed their confirmation to the typical β-sheet state with a positive band at 200 nm and a negative band at 222 nm. However, the Aβ$_{1-42}$/GQDs mixtures that had higher GQDs content (Aβ$_{1-42}$ : GQDs = 1 : 1 and 1 : 5) still remained in the β-turn states (figure 2e,f). Moreover, longer incubating time had little influence on the confirmations of pure Aβ$_{1-42}$ peptide and low GQDs containing mixtures. Additionally, the confirmation of the Aβ$_{1-42}$/GQDs mixture with the mass ratio of 1 : 1 and 1 : 5 showed little changes when the confirmation became to β-turn with the lapse of time.

To investigate the influences of GQDs in the process of Aβ$_{1-42}$ aggregation, the morphologies of the Aβ$_{1-42}$/GQDs mixture with low (Aβ$_{1-42}$ : GQDs = 100 : 1) and high (Aβ$_{1-42}$ : GQDs = 1 : 1) content of GQDs were performed by TEM and the results revealed significant differences as shown in figure 3. With low content of GQDs, the mixture presented the distinct typical fibrils structure (figure 3a),

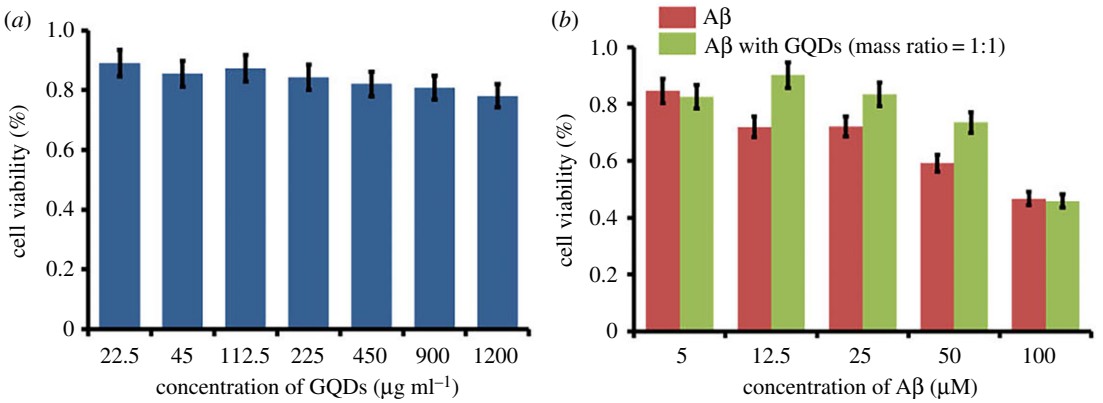

**Figure 4.** (*a*) Cytotoxicity of GQDs and (*b*) cytotoxicity of Aβ$_{1-42}$ peptide and Aβ$_{1-42}$/GQDs mixture with the mass ratio of 1 : 1.

while incubated with high content of GQDs, the typical fibrils disappeared, instead forming a new hybrid network structure (figure 3*b*). Based on CD results, this typical hybrid network structure should also be different from typical fibrils in the secondary structure by containing little β-sheet. CD spectra give us the information that the critical mass ratio of GQDs and Aβ$_{1-42}$ peptide was 1 : 1. Lower content of GQDs had little effect on the confirmation changes of Aβ$_{1-42}$ peptide. If the content of GQDs was enough, GQDs could regulate the confirmation of Aβ$_{1-42}$ peptide into the β-turn state and prevent the confirmation further transforming to the β-sheet state.

## 3.3. GQDs reduced the cytotoxicity induced by Aβ$_{1-42}$

In addition, the cytotoxicity of GQDs and Aβ$_{1-42}$/GQDs mixture against SH-SY5Y cells was evaluated by MTT assay to investigate the safety of GQDs and the changes that GQDs brought to Aβ$_{1-42}$ peptide in biology aspect. MTT assay based on succinic dehydrogenase activity in mitochondria was used to investigate the cell viabilities of SH-SY5Y cells in the presence of GQDs or Aβ$_{1-42}$/GQDs mixture. GQDs were demonstrated to be a biocompatible material with slight toxicity to SH-SY5Y cells as shown in figure 4*a*. SH-SY5Y exhibited high cell viability up to 89% when the concentration of GQDs was 22.5 μg ml$^{-1}$. Even 78% cells were still alive when the concentration increased to 1200 μg ml$^{-1}$, which implied that GQDs could keep a relatively good biocompatibility from low to high dosage. What is more, the toxicity of Aβ$_{1-42}$ peptide to SH-SY5Y cells was greatly reduced in the presence of GQDs. With the addition of Aβ$_{1-42}$ and GQDs mixture (1 : 1 mass ratio), the cell viabilities of SH-SY5Y were promoted significantly when the concentration of Aβ$_{1-42}$ peptide was located in the range of 12.5–50 μM (figure 4*b*), which was associated with the previous reports [7]. However, when the peptide concentration was decreased to 5 μM or increased to 100 μM, the effect of GQDs was almost neglected due to the indistinct toxicity deriving from the slim concentration and the non-affected peptide at high concentration. The possible reason for GQDs to reduce the cell toxicity of Aβ$_{1-42}$ peptide might owe to the structure changes of Aβ$_{1-42}$ peptide induced by GQDs as discussed in CD results. Overall, the results in MTT assay provide an opportunity to apply GQDs in a biological system due to its native low cytotoxicity, detection capability and the abilities to change peptide toxicity.

## 3.4. Detection of the regulatory mechanism of GQDs in the Aβ$_{1-42}$ aggregation

The regulatory mechanism of GQDs in the Aβ$_{1-42}$ aggregation is another critical question. Therefore, a series of contrast experiments were performed to understand the modulation mechanism of GQDs in the process of Aβ$_{1-42}$ aggregation.

DLS is a powerful tool to monitor particle size evolutions. In the experiment, native Aβ$_{1-42}$ and Aβ$_{1-42}$/GQDs mixtures were incubated at 37°C for 72 h to obtain Aβ$_{1-42}$ fibrils and mature Aβ$_{1-42}$/GQDs mixture. As shown in figure 5*a*,*b*, the size of GQDs or Aβ$_{1-42}$ peptide ranged from 5–30 nm in DLS histograms. While the size distribution of the Aβ$_{1-42}$/GQDs mixtures was strikingly grown to 100–900 nm, the particles with size less than 50 nm disappeared (figure 5*d*), indicating a co-assembly between GQDs and Aβ$_{1-42}$ peptide. As in the previous report, the electrostatic interaction played an important role in the interaction between Aβ$_{1-42}$ peptide and other electronegativity molecules [12]. The characterizations of GQDs demonstrated that GQDs possessed abundant oxygen-containing groups including carboxyl group,

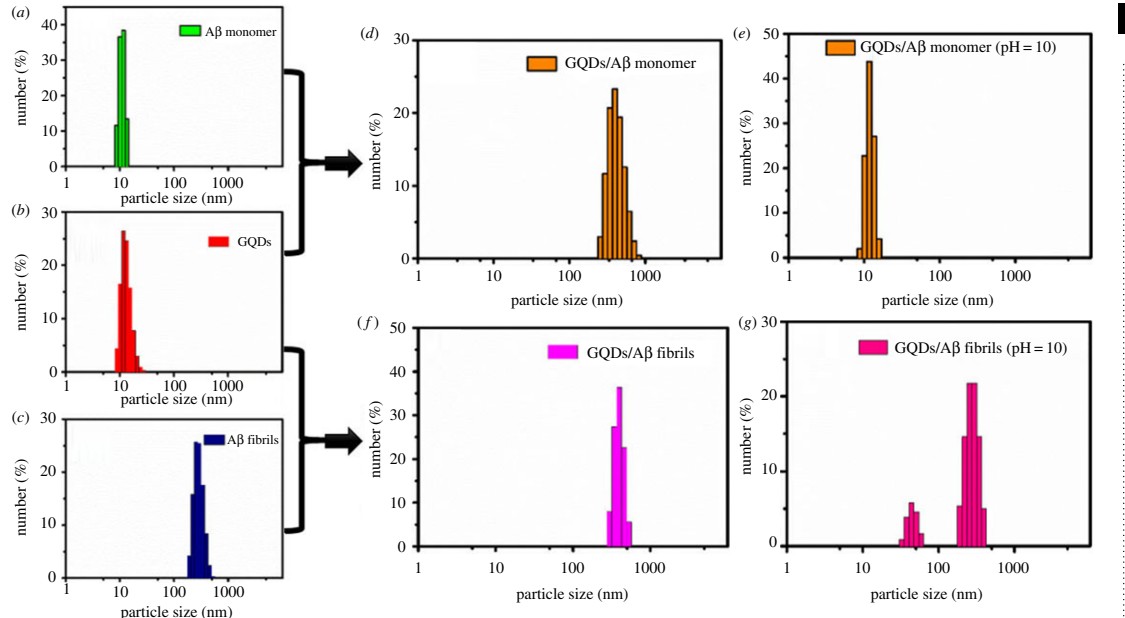

**Figure 5.** DLS spectra of (*a*) Aβ$_{1-42}$ peptide, (*b*) GQDs, (*c*) Aβ$_{1-42}$ fibrils, (*d*) Aβ$_{1-42}$ peptide/GQDs with mass ratio of 1:1, (*e*) Aβ$_{1-42}$ peptide/GQDs with mass ratio of 1:1 (pH = 10), (*f*) Aβ$_{1-42}$ fibrils/GQDs with mass ratio of 1:1, (*g*) Aβ$_{1-42}$ fibrils/GQDs with mass ratio of 1:1 (pH = 10).

which is electronegative. Therefore, an increased pH value of above mixtures was administrated which could be used as a useful method to break the electrostatic interaction. As a result, the particle size of the mixture decreased greatly from 100–900 nm to 5–20 nm (figure 5*e*), which implied that a decomposition reaction occurred upon the pH jump. The resulting particle size was also close to the original size distribution of GQDs and free Aβ$_{1-42}$ peptide; thus, the particles dispersed in this system should be the dissociative GQDs and Aβ$_{1-42}$ peptide.

To further investigate whether the GQDs could disaggregate the mature Aβ$_{1-42}$ fibrils, the size evolution of mature Aβ$_{1-42}$ fibrils was also measured in the presence of GQDs. As the results shown in figure 5*c*, the particle size of Aβ$_{1-42}$ fibrils was measured to be 150–550 nm and no obvious variations were observed after being mixed with GQDs. In addition, the size distribution of GQDs within 5–30 nm did not appear after adding the GQDs into the Aβ$_{1-42}$ fibrils (figure 5*f*), which might result from the adsorption of GQDs on the Aβ$_{1-42}$ fibrils. However, two distribution intervals appeared in histograms when the pH value was adjusted to 10 in figure 5*e*. The particle size distribution within 100–500 nm was consistent with the particle size of mature Aβ$_{1-42}$ fibrils, indicating the negligible changes of the mature fibrils in the presence of GQDs upon the change of pH value. Besides, the size distribution within 20–60 nm was a little bigger than the free GQDs which might be due to the aggregation of GQDs with other components in the mixture, such as other GQDs formulation, unassembled peptide or dissociative oligomers.

Visual evidence was also provided by TEM images to confirm the morphology changes of the aggregates detected in DLS assay. As shown in the results, figure 6*a*–*c* exhibited the morphologies of native Aβ$_{1-42}$ peptide, GQDs and fibrillar Aβ$_{1-42}$, respectively. The rest of the images were the resulting morphologies after being treated with the same procedure as DLS tests. A distinct disordered network structure appeared when the mixture of the GQDs and Aβ$_{1-42}$ peptide reacted for 72 h (figure 6*d*). While the pH value changed to 10, the cross-like structure disappeared as shown in figure 6*e*, instead of small particles scattering on the ground. In the image of GQDs and fibrillar Aβ$_{1-42}$ mixture, only the cross-like structure was observed and the GQDs were not observed in the background (figure 6*f*). This result implied that the GQDs could absorb to the fibrillar Aβ$_{1-42}$. When the pH value of the system was adjusted to 10, the mixture of GQDs and fibrillar Aβ$_{1-42}$ displayed morphology with separated linear fibrils and small particles (figure 6*g*). All the morphology features shown in TEM images can be well coincident with the data in DLS.

Consequently, these results suggested that the mixture of GQDs and Aβ$_{1-42}$ peptide could form aggregates with particle sizes of 100–900 nm. This assembly process was mainly driven by the electrostatic interaction, resulting in a unstable cross-like network structure, which would totally collapse when the environment slightly changed into alkaline. However, when the Aβ$_{1-42}$ peptide

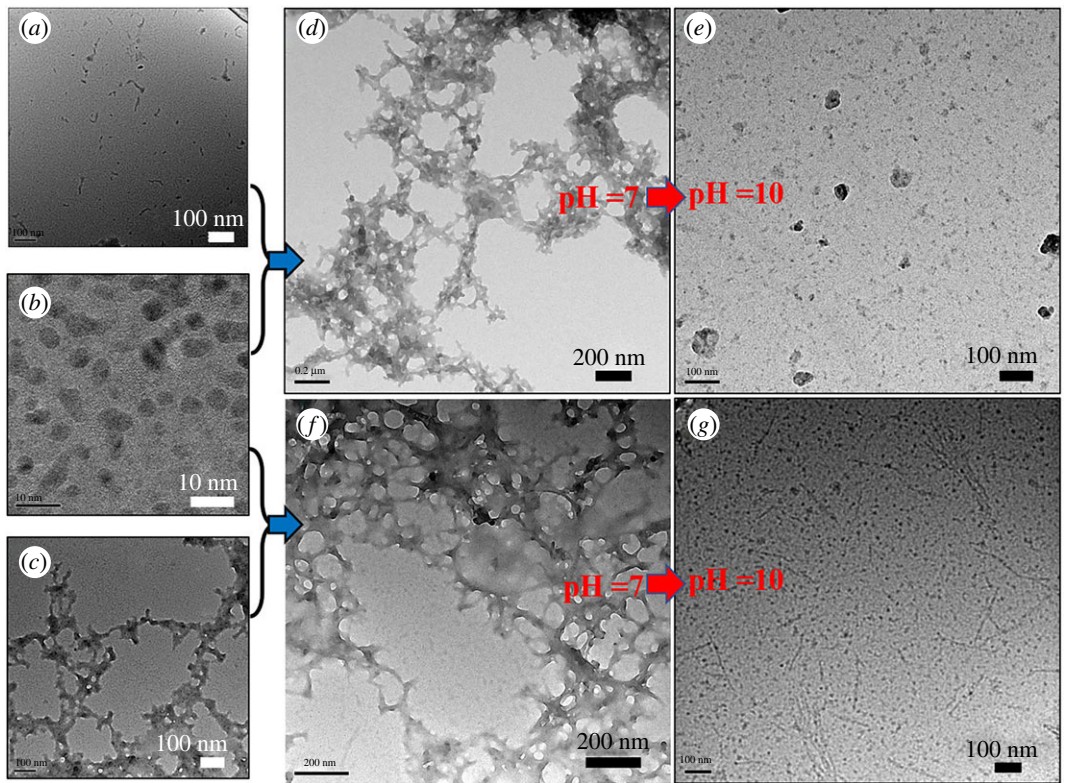

**Figure 6.** TEM images of (a) Aβ$_{1-42}$ peptide; (b) GQDs; (c) Aβ$_{1-42}$ fibrils; (d) Aβ$_{1-42}$ peptide/GQDs with mass ratio of 1:1 in PBS (pH = 7); (e) Aβ$_{1-42}$ peptide/GQDs with mass ratio of 1:1 in PBS buffer, the pH was adjusted to 10 by NaOH; (f) Aβ$_{1-42}$ fibrils/GQDs with mass ratio of 1:1 in PBS (pH = 7); (g) Aβ$_{1-42}$ fibrils/GQDs with mass ratio of 1:1 in PBS buffer, the pH was adjusted to 10 by NaOH.

assembled into mature fibrils, the GQDs could absorb on the fibrils, then the Aβ$_{1-42}$ fibrils would maintain when the pH value jumped to 10. Thus, the network structure formed by GQDs and Aβ$_{1-42}$ peptide should not contain the mature Aβ$_{1-42}$ fibrils. The GQDs not only have the tendency to combine with Aβ$_{1-42}$ peptide but also can prevent the conventional fibrosis process by this combination effect. It is worth mentioning that GQDs could not disintegrate the fibrillar Aβ$_{1-42}$ that has already formed.

## 3.5. Proposed mechanism

Based on the above analysis as well as CD spectra, DLS and TEM images, a possible mechanism was proposed in figure 7. If a sufficient quantity of GQDs was mixed with Aβ$_{1-42}$ peptide (peptide/fibrils) and incubated for an extended period, the possible reaction processes were shown in figure 7. For the mixture of GQDs and Aβ$_{1-42}$ peptide, they were prone to assembly into an unstable cross-like structure through electrostatic interaction, $\pi-\pi$ stacking and hydrogen bonding, in which the electrostatic interaction is the main interaction. Because of the strong interaction between Aβ$_{1-42}$ and GQDs, the secondary structure of the Aβ$_{1-42}$ aggregates could maintain the β-turn state in the experimental period according to the results of CD spectra and cell cytotoxicity. Subsequent pH jump will raise the electronegativity of GQDs and Aβ$_{1-42}$ peptide by the deprotonation of the carboxyl groups. As a result, electrostatic repulsive force dominated the interactions and impelled the disassembly of the Aβ$_{1-42}$/GQDs co-assembled structure. For the mixture of GQDs and Aβ$_{1-42}$ mature fibrils, the GQDs only absorbed on the surface of fibrils without damaging the performed fibrils. The adjustment of the pH value could only damage the absorption force between GQDs and Aβ$_{1-42}$ fibrils but could not damage the fibrous structure.

## 4. Conclusion

According to the above research and discussions, the feasibility of using GQDs to regulate the aggregation of Aβ$_{1-42}$ was demonstrated. The critical mass ratio of Aβ$_{1-42}$ and GQDs was 1:1.

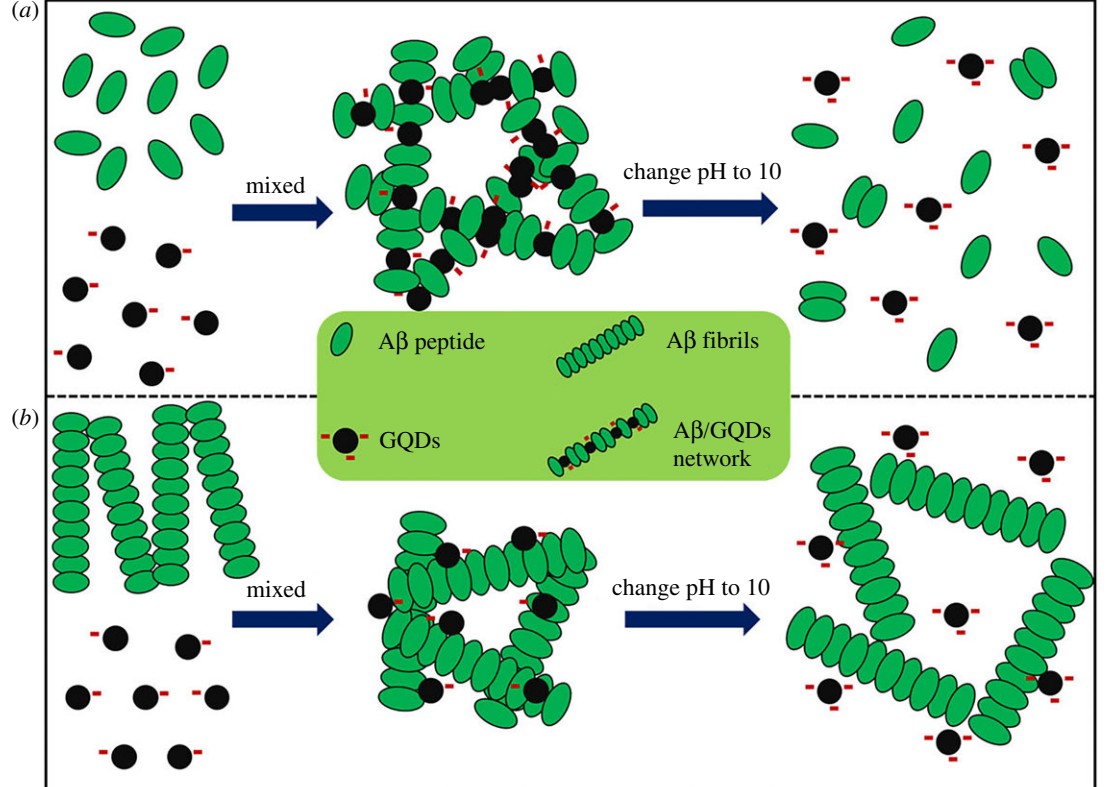

**Figure 7.** Proposed mechanistic diagrams for GQDs interact with $A\beta_{1-42}$ peptide (*a*) and fibrils (*b*), and their structure changes after adjusting pH value to 10.

What is more, GQDs could protect $A\beta_{1-42}$ from being aggregated into big fibrils and maintain the aggregates in the nontoxic state but could not break the fibrils that had already formed. It is worth pointing out that the way that GQDs inhibited $A\beta_{1-42}$ aggregation was forming a hybrid network structure by the strong interaction between GQDs and native $A\beta_{1-42}$ peptide instead of keeping native $A\beta_{1-42}$ freedom in solution. Additionally, great changes to the sizes and morphologies in the mixture upon pH jump strongly demonstrated the important role that the electrostatic interaction played in the interaction of GQDs and $A\beta_{1-42}$. Moreover, *in vitro* tests produced credible evidence of the positive regulation of $A\beta_{1-42}$ by GQDs. As a consequence, GQDs can be widely used as a functional nanomaterial in the area of protein amyloidogenesis.

Data accessibility. The datasets supporting this article have been uploaded as part of the electronic supplementary material.

Authors' contributions. C.L. and H.H. carried out the experiment, collected and analysed the data and drafted the manuscript. L.M. and X.F. revised the manuscript. C.W. and Y.Y. conceived and supervised the study. All authors gave final approval for publication.

Competing interests. We declare we have no competing interests.

Funding. This work was supported by the National Key Research and Development Program of China (2016YFF0203803), the National Natural Science Foundation of China (no. 21773042) and the Beijing Natural Science Foundation (no. 2162044).

Acknowledgements. We thank Beijing Regional Center of Nanoscience Instrument for their assistance in characterizations of dynamic light scattering, circular dichroism, transmission electron microscopy studies, X-ray photoelectron spectra, Raman spectra and Fourier-transform infrared spectra.

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
