## [Reviewer comments · Royal Society Open Science]

Review History

RSOS-190271.R0 (Original submission)

Review form: Reviewer 1 (Nidhi Gour)

Is the manuscript scientifically sound in its present form?

Yes

Are the interpretations and conclusions justified by the results?

Yes

Is the language acceptable?

Yes

Is it clear how to access all supporting data?

Yes

Do you have any ethical concerns with this paper?

No

Have you any concerns about statistical analyses in this paper?

No

Recommendation?

Accept with minor revision (please list in comments)

Comments to the Author(s)

The manuscript by Liu et al. is a nice piece of research and should be accepted. I recommend certain minor revisions with the language used and it would be nice if help of native english speaker could be sought. In general, the modulator effect of GQDs on A β 1-42 aggregation is very well explained and well-characterized by the techniques like TEM, XPS, FTIR and CD. Finally, MTT assay is also done which proves GQD are bio-compatible and can serve as good therapeutic tool against Alzheimer's.

Review form: Reviewer 2 (Zhujun Chen)

Is the manuscript scientifically sound in its present form?

Yes

Are the interpretations and conclusions justified by the results?

Yes

Is the language acceptable?

Yes

Is it clear how to access all supporting data?

Yes

Do you have any ethical concerns with this paper?

No

Have you any concerns about statistical analyses in this paper?

No

Recommendation?

Accept with minor revision (please list in comments)

Comments to the Author(s)

In this manuscript, authors studied the regulatory effects and mechanism of graphene quantum dots (GQDs) on 1-42 β -amyloid (A β 1-42) aggregation. Based on their study, the mechanism of the regulatory effects of GQDs on the A β 1-42 aggregation was proposed. This manuscript is well suitable for publication in Royal Society Open Science after minor revision.

1. It's better to describe more clearly (with more detail) about the experimental procedures, especially concerning the fiber formation and inhibition process (including pH adjustment).
2. Fig. 3 indicated the time mass ratio of GQDs and A β 1-42 peptide was 1: 1. How to determine the ratio?
3. Some typo or format problems: H2SO4 and HNO3, etc.

Decision letter (RSOS-190271.R0)

24-Apr-2019

Dear Mr Liu:

Title: Modulation of β -Amyloid Aggregation by Graphene Quantum Dots
Manuscript ID: RSOS-190271

Thank you for submitting the above manuscript to Royal Society Open Science. On behalf of the Editors and the Royal Society of Chemistry, I am pleased to inform you that your manuscript will be accepted for publication in Royal Society Open Science subject to minor revision in accordance with the referee suggestions. Please find the reviewers' comments at the end of this email.

The reviewers and handling editors have recommended publication, but also suggest some minor revisions to your manuscript. Therefore, I invite you to respond to the comments and revise your manuscript.

Please also include the following statements alongside the other end statements. As we cannot publish your manuscript without these end statements included, if you feel that a given heading is not relevant to your paper, please nevertheless include the heading and explicitly state that it is not relevant to your work. We have included a screenshot example of the end statements for reference.

- Ethics statement

Please clarify whether you received ethical approval from a local ethics committee to carry out your study. If so please include details of this, including the name of the committee that gave consent in a Research Ethics section after your main text. Please also clarify whether you received informed consent for the participants to participate in the study and state this in your Research Ethics section.

OR

Please clarify whether you obtained the necessary licences and approvals from your institutional animal ethics committee before conducting your research. Please provide details of these licences and approvals in an Animal Ethics section after your main text.

OR

Please clarify whether you obtained the appropriate permissions and licences to conduct the fieldwork detailed in your study. Please provide details of these in your methods section.

Because the schedule for publication is very tight, it is a condition of publication that you submit the revised version of your manuscript before 03-May-2019. Please note that the revision deadline will expire at 00.00am on this date. If you do not think you will be able to meet this date please let me know immediately.

When submitting your revised manuscript, you will be able to respond to the comments made by the referees and upload a file "Response to Referees" in "Section 6 - File Upload". You can use this to document any changes you make to the original manuscript. In order to expedite the

processing of the revised manuscript, please be as specific as possible in your response to the referees.

Best wishes,
Dr Laura Smith
Publishing Editor, Journals

On behalf of the Subject Editor Professor Anthony Stace and the Associate Editor Professor Claire Carmalt.

RSC Associate Editor:
Comments to the Author:
(There are no comments.)

RSC Subject Editor:
Comments to the Author:

(There are no comments.)

Reviewer comments to Author:

Reviewer: 1

Comments to the Author(s)

The manuscript by Liu et al. is a nice piece of research and should be accepted. I recommend certain minor revisions with the language used and it would be nice if help of native english speaker could be sought. In general, the modulator effect of QDs on A β 1-42 aggregation is very well explained and well-characterized by the techniques like TEM, XPS, FTIR and CD. Finally, MTT assay is also done which proves QD are bio-compatible and can serve as good therapeutic tool against Alzheimer's.

Reviewer: 2

Comments to the Author(s)

In this manuscript, authors studied the regulatory effects and mechanism of graphene quantum dots (GQDs) on 1-42 β -amyloid (A β 1-42) aggregation. Based on their study, the mechanism of the regulatory effects of GQDs on the A β 1-42 aggregation was proposed. This manuscript is well suitable for publication in Royal Society Open Science after minor revision.

1. It's better to describe more clearly (with more detail) about the experimental procedures, especially concerning the fiber formation and inhibition process (including pH adjustment).
2. Fig. 3 indicated the time mass ratio of GQDs and A β 1-42 peptide was 1: 1. How to determine the ratio?
3. Some typo or format problems: H2SO4 and HNO3, etc.

Author's Response to Decision Letter for (RSOS-190271.R0)

See Appendix A.

Decision letter (RSOS-190271.R1)

23-May-2019

Dear Mr Liu:

Title: Modulation of β -Amyloid Aggregation by Graphene Quantum Dots

Manuscript ID: RSOS-190271.R1

I am pleased to inform you that your manuscript is now accepted for publication in Royal Society Open Science. The chemistry content of Royal Society Open Science is published in collaboration with the Royal Society of Chemistry.

In order to raise the profile of your paper once it is published, we can send through a PDF of your paper to selected colleagues. If you wish to take advantage of this, please reply to this email with the name and email addresses of up to 10 people who you feel would wish to read your article.

On behalf of the Editors of Royal Society Open Science and the Royal Society of Chemistry, we look forward to your continued contributions to the Journal.

Best wishes,
Andrew Dunn

On behalf of the Subject Editor Professor Anthony Stace and the Associate Editor .

Appendix A

Professor Yanlian Yang
National Center for Nanoscience and Technology,
University of Chinese Academy of Sciences, Beijing 100190, China.
E-mail: yangyl@nanoctr.cn

Dear Prof. Laura Smith,

We would like to thank you for giving us such an opportunity to revise our manuscript. We have made appreciate modifications on the original manuscript to address the concerns of the reviewers and editors. Here, we attach the revised manuscript entitled “**Modulation of β -Amyloid Aggregation by Graphene Quantum Dots**” (Manuscript ID: **RSOS-190271**) for your consideration for publication as an article in *Royal Society Open Science*. Thank you for your great efforts in obtaining the reviewers’ comments. We also appreciate the critical review and constructive comments of the reviewers to help improve our manuscript. The manuscript has been revised according to the suggestions from the reviewers, and all the revisions have been highlighted in yellow for clarity. The responses to the reviewers’ comments with a point-by-point list of the changes are provided in separate pages.

Thanks a lot for your consideration.

Sincerely yours,

Yanlian Yang, Professor

National Center for Nanoscience and Technology

No. 11, Beiyitiao Zhongguancun, Beijing, 100190, China

E-mail: yangyl@nanoctr.cn

Fax: +86 10 6265 6765

Tel: +86 10 8254 5559

Response to the Reviewer #1's comments

We would like to express our sincere thanks to your positive comments, which are quite helpful to improve the quality of our article. We have revised our paper according to your comments. The main revisions are listed as follows:

Point 1:

I recommend certain minor revisions with the language used and it would be nice if help of native English speaker could be sought.

Author's Response:

Thanks a lot for your kind advice. We have modified the article carefully in terms of language. The changes have been highlighted in yellow.

(lines 25, 33-34 in page 1, line 2-5, 11-12, 34-35 and 47-48 in page 2, etc.)

Response to the Reviewer #2's comments

We would like to express our sincere thanks to your positive comments. Those comments are all valuable for improving our paper. We have revised the manuscript point-by-point according to your comments. The main revisions are listed as follows:

Point 1:

It's better to describe more clearly (with more detail) about the experimental procedures, especially concerning the fiber formation and inhibition process (including pH adjustment).

Author's Response:

Thanks a lot for your kindly advices. We have described the experimental procedure more clearly according to your suggestion.

The fiber formation of A β ₁₋₄₂ was prepared by incubating the peptide solution at 37 °C for 72 h. In the brief, A β ₁₋₄₂ powder was dissolved in 1,1,1,3,3,3-hexafluoro-2-propanol (HFIP) at a concentration of 1 mg mL⁻¹, and further incubated at 37 °C for 6 h. Then the dissolved A β ₁₋₄₂ solution was separated into 100 μ L per tube and dried under nitrogen gas. Afterwards, the dried A β ₁₋₄₂ peptide was re-dissolved in PBS buffer at a concentration of 20 μ M with 10 μ L DMSO as cosolvent. The A β ₁₋₄₂ peptide was put in a constant temperature shaker at 37 °C and incubated for 72 h to form the mature fibril.

In the inhibition process of A β ₁₋₄₂ peptide aggregation by GQDs, the A β ₁₋₄₂ peptide was dissolved into PBS at a concentration of 50 μ M in the presence of 10 μ L DMSO. The GQDs was lyophilized and re-dissolved into PBS at a concentration of 10 mg/mL. After that, a certain amount of A β ₁₋₄₂ peptide and GQDs with the mass ratio of 100: 1, 10: 1, 1: 1, 1: 5 were mixed and PBS buffer was added into the mixture to adjust the final concentration of A β ₁₋₄₂ peptide at 20 μ M. The free A β ₁₋₄₂ peptide with the concentration of 20 μ M was prepared as control. After being incubated for 72 h, the inhibitory effect of GQDs in A β ₁₋₄₂ peptide aggregation was investigated by TEM, DLS and CD.

The same procedure was used to investigated the regulatory effect of GQDs in

A β ₁₋₄₂ peptide aggregation at pH 10. When the mixture prepared according to the method mentioned above was incubated for 72 h, 0.1 M NaOH was added into the mixture with 5 μ L per drop, the pH of the system was monitored by a mini pH detector. The mixture was incubated for another 72 h in a constant temperature shaker at 37 °C before detected by TEM, DLS when the pH was adjusted to 10.

(line 112-13, 6-17 in page 4)

Point 2:

Fig. 3 indicated the mass ratio of GQDs and A β ₁₋₄₂ peptide was 1: 1. How to determine the ratio?

Author's Response:

Thanks a lot for your suggestion. We are so sorry for missing the relevant information of the method to determine the mass ration of GQDs and A β ₁₋₄₂ peptide in experimental section.

Briefly, A β ₁₋₄₂ powder was dissolved in 1,1,1,3,3,3-hexafluoro-2-propanol (HFIP) at a concentration of 1 mg mL⁻¹, and further incubated at 37 °C for 6 h. Then the dissolved A β ₁₋₄₂ solution was separated into 100 μ L per tube and dried under nitrogen gas. Afterwards, the dried A β ₁₋₄₂ peptide was re-dissolved in PBS buffer at a concentration of 50 μ M with 10 μ L DMSO as cosolvent. The GQDs was lyophilized and re-dissolved into PBS at a concentration of 10 mg/mL. then the GQDs was added into the A β ₁₋₄₂ peptide solution with the mass ratio of 1: 1. After that, additional PBS was added into the mixture to adjust the final concentration of A β ₁₋₄₂ peptide at 20 μ M. Last, the morphology of the mixture was observed by TEM after being incubated for 72 h.

Point 3:

Some typo or format problems: H2SO4 and HNO3, etc.

Author's Response:

Thanks a lot for your suggestion. We are sorry for the careless mistakes in the manuscript. We have checked the whole article and all the similar mistakes have been corrected.

(line 39 in page 4)